# HiDet: Hierarchical AI Text Detection via Coarse Filtering and Multi-Grained Contrastive Learning

## Abstract

Humanizing methods of AI-generated texts are emerging, which leads to severe performance degradation of current AI text detectors. Most existing detectors are struggling to ensure consistent performance in the huge span from detecting simple AI texts to detecting AI texts humanized in various ways. In addition, the monolithic threshold-based scoring mechanism they rely on is vulnerable, and some humanized AI texts can escape sanction after a single detection. Targetedly, we propose **HiDet**, which contains coarse module and subdivision module to give AI texts a double check. We decouple the complete detection process into simple sample detection and difficult sample detection, the coarse module of HiDet filters out simple samples, while hard-to-detect samples like humanized AI texts will be carefully discriminated through the subdivision module which applies a multi-grained contrastive learning strategy. This hierarchical framework makes up for the loophole that humanized AI texts can successfully escape the traditional detector after a single detection, and shows excellent robustness in the task of detecting humanized AI texts. Meanwhile, our framework is flexiable, the subdivision module can be deployed separately on the existing detector as a plug-and-play patch to tremendously improve their performance when facing large-scale humanized AI texts. We hope our work can inspire new sparks in the field of AI-generated text detection, codes and datasets will be open soon.

## 1 Introduction

The explosive rise of LLMs (Claude AI, 2024; DeepSeek-AI, 2025; Gemini, 2024) makes it easy for people to get AI-generated texts. However, new problems emerge when the text generation function of LLMs facilitates people's work and life. Researchers have demonstrated various malicious applications of LLMs, including academic fraud (Perkins, 2023), spam generation, and false information dissemination (Hazell, 2023; Weidinger et al., 2022). For instance, by leveraging Chat-GPT's powerful writing capabilities, attackers create a large number of automated bots on social networks, successfully manipulating people's political choices during elections (Solaiman et al., 2019b; Goldstein et al., 2023). In addition, many studies have pointed out that students complete the entire content of papers and assignments through LLMs, which leads to the spread of academic misconduct (Bhaskar and Rana, 2024; Mitchell, 2022). In order to prevent AI-generated texts from leaning into the wrong direction, AI text detectors come into being to correct the development of LLMs. Existing detectors include those based on statistics and mathematics (Mitchell et al., 2023; Tian and Cui, 2023), watermarks (Gu et al., 2022; Kirchenbauer et al., 2023), classifiers (Guo et al., 2023; Wang et al., 2023) and other frameworks[1], they together establish defence against harmful usage of AI texts.

However, the defence they build is not impenetrable. A great number of humanizing methods created by attackers with ulterior motives are attempting to break the defence and researchers have conducted solid work to point out that current AI text detectors are vulnerable (Zhou et al., 2024; Dugan et al., 2024; Krishna et al., 2024; Liu et al., 2024b; Huang et al., 2024; Wang et al., 2024). As an illustration, Dugan (Dugan et al., 2024) designs a variety of perturbations, finding that with only 5% of the text

---

[1]Detailed introduction for other detectors is in Appendix F.

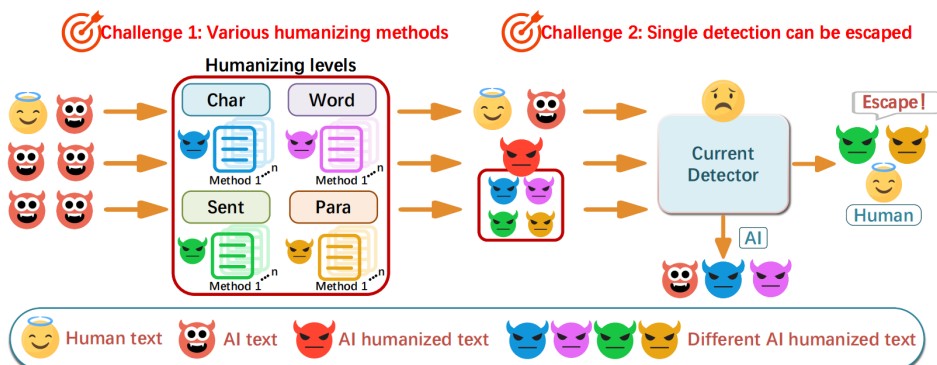

Figure 1: Illustration of two challenges: (1) The difficulty of catching AI texts humanized in various ways and different levels. (2) The difficulty of catching AI texts through one detection.

content being modified, the detection accuracy drops by an average of 37.2%. No doubt, current detectors are facing challenges.In our view, two main critical challenges should be sovled to further break the bottleneck and improve AI detectors' robustness, illustrating in Figure 1.

For the first challenge, various humanizing methods disguise AI texts and are making them more difficult to detect. Humanizing methods, which can be roughly divided into four levels: character, word, sentence, and paragraph (Zhou et al., 2024), include simulating human spelling errors, word replacement, sentence back translation and new methods are appearing wildly. In order to address this challenge, researchers have actively explored and tentatively proposed many robust detector (Hans et al., 2024; Hu et al., 2023; Tian and Cui, 2023). While they do show resistance to certain attacks, some of them still fail to defend the wide range attack from char level to paragraph level (Dugan et al., 2024; Liu et al., 2024b).

The second challenge is that single detection often struggles to effectively manage diverse and complex scenarios (Fariello et al., 2024). To illustrate, a great number of detectors are based on a monolithic threshold-based scoring mechanism. They compare the score of the input text with a certain threshold and directly reach a conclusion that whether the text is generated by human or AI. As the current LLMs are getting more skilled at simulating human writing (Chang et al., 2024), and the humanizing methods for AI texts are becoming more and more diverse (Nguyen-Son et al., 2022), the difficulty of catching all AI texts through one detection and one method is rising sharply. Faced with this dilemma, we engage in proactive thinking: Is it possible to catch the AI texts that escaped detection step by step through multiple rounds of detection?

To solve the two challenges mentioned above, we propose HiDet, a hierarchical AI text detection framework with coarse filtering and multi-grained contrastive learning. Detailedly, we break the whole detection process into simple sample detection and difficult sample detection, and design two modules from coarse to fine accordingly. Coarse module aims to give texts a rough filtering, when the scores of texts are within the AI category, these texts are unhumanized and easily detectable AI texts so the detection results are directly reached. When the texts' scores are within the human category, the texts may not only be human written, but also humanized AI texts, thus will enter the subdivision module for fine-grained detection. In the subdivision module, we divide the texts into four granularities: (1) AI texts humanized in the same method, (2) AI texts with humanized methods of the same level, (3) humanized AI texts and original AI texts, (4) human texts and AI texts, and our multi-grained contrastive Learning is introduced to further dig into the nuances of AI texts. Through our framework, we not only catch samples that escape detection in one round by giving them a second check, but also effectively defend against humanized AI text attacks with a subdivision module designed specifically for humanized samples. The two modules each perform their respective functions and together build a wall to resist AI text invasion.

We achieve State-Of-The-Art (SOTA) performance in not only detecting original AI texts but also humanized AI texts through this coarse-to-fine detection framework, which indicates HiDet's great robustness in defending various attack. What's more, as our subdivision module are specially designed for humanized AI texts, we add it to the current AI text detectors, successfully alleviating their stress

under large scale humanized AI texts attack. This result further shows that our work is flexiable and has great practical significance: The subdivision module can serve as a possible plug-and-play patch to further improve some current detectors' performance in detecting humanized AI texts.

In short, our work is multifaceted and can be summarized as follows:

- We propose a new framework for training detectors, with coarse module targeting at detecting original AI texts and subdivison module focusing on humanized AI texts, providing ideas for detector designers to resist the humanizing attacks.

- We design multi-grained contrastive learning in subdivision module to achieve distinction between human texts and AI texts humanized at different levels, which can also be used as a plug-and-play patch to improve the robustness of existing detectors.

- We conduct extensive experiments on two tasks of detecting original AI texts and humanized AI texts, results show that our detector has achieved SOTA performance. We also apply our subdivision module to some detectors and achieve significant improvements in their performance.

## 2 RELATED WORKS

**Multi-stage detection.** The advantage of multi-stage detection is that it can subdivide the detection process and make it more granular. Jiang (Jiang et al., 2024) and Feng (Feng et al., 2024) divide the software vulnerability detection process into coarse-grained detection and fine-grained positioning. Cao (Cao et al., 2024) proposes a two-stage tax evasion detection. Concone (Concone et al., 2023) uses low-cost computing features to quickly filter easy-to-classify samples in spam detection and only performs fine-grained distinction on difficult samples. Duan (Duan et al., 2020) uses two classifiers to coperform object detection. The multi-stage detection framework is also widely used in detecting video event abnormality, network attack, conversation emotion, false information, and so on (Wang et al., 2018; 2025; Li et al., 2024; Lin et al., 2024; Li et al., 2021).

**Humanizing methods of AI texts.** Many researchers have pointed out the vulnerability of current detectors, indicating even a small perturbation attack can cause the performance of the detector to drop sharply (Zhou et al., 2024; Dugan et al., 2024; Krishna et al., 2024; Liu et al., 2024b; Huang et al., 2024; Wang et al., 2024). Specifically, Liu (Liu et al., 2024b) points out that DetectGPT relies on the threshold setting of the logit regression module, which is sensitive to the detection results, and perturbations of deletion, duplication, insertion, replacement imposed on the test data cause the performance of detector to drop significantly; Dugan (Dugan et al., 2024) designs a variety of perturbations such as local vocabulary replacement, syntactic structure adjustment, semantic preservation rewriting, finding that with only 5% of the text content being modified, the detection accuracy drops by an average of 37.2%; Zhou (Zhou et al., 2024) concludes humanizing methods into four levels of character, word, sentence, and paragraph on a variety of research (we apply this idea), pointing out that current detectors need to be trained with adversarial texts. Their works show that lacerating the mask of humanized texts is an urgent affair which needs high attention.

## 3 MODEL AND METHODOLOGY

### 3.1 FRAMEWORK OVERVIEW

There are two tasks for AI text detectors. Task 1 only involves human texts and original AI texts thus are relatively simple. Task 2 takes humanized AI texts into consideration and is more realistic, we choose 19 humanizing methods and summarize them into four levels, detailed introduction is in Appendix C. We define the two tasks below:

$$\text{Task 1}: \quad \text{Detecting human texts and AI texts}$$
$$\text{Task 2}: \quad \text{Detecting human texts, AI texts and humanized AI texts}$$

Our goal is to ultimately determine whether sample $s$ is AI generated or not. For Task 1, coarse module will complete most of the work ,very few AI texts will bypass it and subdivision module

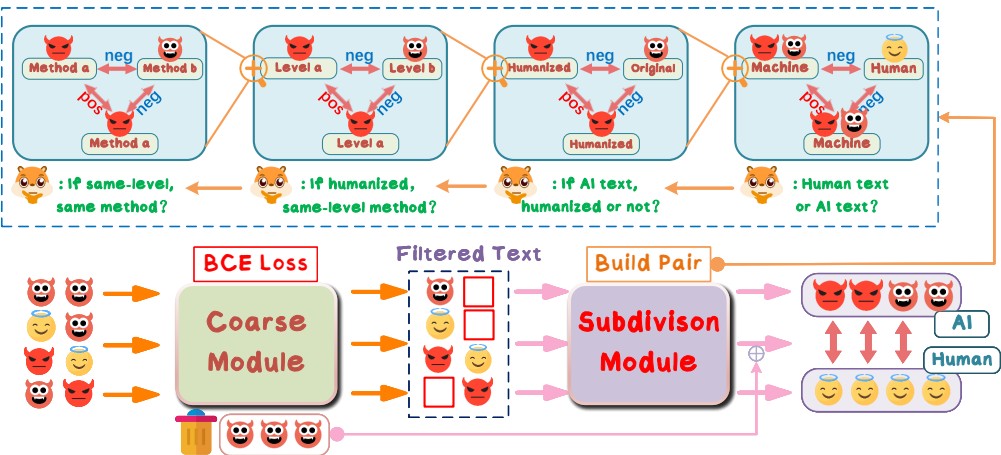

Figure 2: HiDet's framework overview. Coarse module filters easy-to-detect samples, while difficult samples will be detected again by subdivision module with multi-grained contrastive learning.

will detect them again. For Task 2, the process is similar. Sample $s$ will first pass through the coarse module, the final text's score is compared with the threshold and if it is in the AI range, then the conclusion that $s$ is AI generated is directly reached, indicating that some simple AI samples are successfully filtered by the first module. If the text is judged to be human written, the text may be real human text or AI written text with humanizing methods. Next, it will undergo the subdivision module with multi-grained contrastive learning for fine-grained differentiation, ultimately determining whether $s$ is indeed a humanized AI text.

## 3.2 COARSE MODULE

The coarse module is a traditional classifier-based detector. Its purpose is to perform coarse-grained screening of input batch texts. This module can also be used as a representative of the current detector that has not been specially trained with text perturbations. Training on unhumanized datasets, a good coarse module should be able to filter out most of the original AI texts that have not been humanized. Its loss function is as follows,in which $x$ represents true label, $y$ represents predicted label:

$$\mathcal{L}_{ce} = -\frac{1}{N}\sum_{i=1}^{N} x_i \cdot log(y_i) + (1 - x_i) \cdot log(1 - y_i), \tag{1}$$

For input $s \in S$, the text encoding $\Phi(s)$, the output is:

$$Prediction(s) = \begin{cases} human, score(\Phi(s)) > threshold \\ machine, scores(\Phi(s)) < threshold \end{cases} \tag{2}$$

The classification threshold is determined by experimental parameter adjustment, detailed in Section 4.2.

## 3.3 SUBDIVISION MODULE

As elaborated in section 3.1 and Figure 2, there are four granularities of multi-grained contrastive learning in the subdivision module: AI texts using the same humanizing method and AI texts with different humanizing methods, AI texts using the same level of humanizing method and AI texts with different levels of humanizing methods, humanized AI texts and original AI texts, human texts and AI texts. Given a text label tuple $(p, q, r, l)$, where $p$ represents the humanizing method used, $q$ represents the level of the humanizing method, $r$ represents whether humanizing method is used or not, and $l$ represents the source of the sample (human or AI). Given text sample $s_i$ and $s_j$, text

encoder $\Phi$, the cosine similarity between two samples is defined as:

$$Sim(\Phi(s_i), \Phi(s_j)) = \frac{\Phi(s_i) \cdot \Phi(s_j)}{||\Phi(s_i)|| \cdot ||\Phi(s_j)||} \quad (3)$$

we have the cosine similarity constraints at different granularities:

$$\begin{cases} Sim(\Phi(s_1), \Phi(s_2)) > Sim(\Phi(s_1), \Phi(s_3)), p(s_1) = p(s_2), p(s_1) \neq p(s_3) \\ Sim(\Phi(s_4), \Phi(s_5)) > Sim(\Phi(s_4), \Phi(s_6)), q(s_4) = q(s_5), q(s_4) \neq q(s_6) \\ Sim(\Phi(s_7), \Phi(s_8)) > Sim(\Phi(s_7), \Phi(s_9)), r(s_7) = r(s_8), r(s_7) \neq r(s_9) \\ Sim(\Phi(s_{10}), \Phi(s_{11})) > Sim(\Phi(s_{10}), \Phi(s_{12})), l(s_{10}) = l(s_{11}), l(s_{10}) \neq l(s_{12}) \end{cases} \quad (4)$$

From top to bottom, the constraints aim to ensure that the similarity of AI texts humanized by the same method is greater than that of AI texts humanized by different methods; the similarity of AI texts humanized to the same level is greater than that of AI texts humanized to different levels; the similarity of AI texts that both either use or do not use humanizing strategy is greater than that between one that uses it and one that does not; and finally, the similarity of texts from the same source is greater than that of texts from different sources.

For contrastive learning at a specific granularity, we use the contrastive loss based on Guo's framework (Guo et al., 2024), which takes the form of a negative logarithmic aggregation function, for label $p$, we have the loss expression Eq. 5, in which $N_m$ represents the number of AI samples in a batch, $z_{i,\mathrm{p}}^+$ is the average similarity between sample i and the positive samples, $z_{i,\mathrm{p}}^-$ is the average similarity between sample i and the negative samples, $\tau$ is the temperature coefficient:

$$\mathcal{L}_{\mathrm{p}} = \frac{1}{N_m} \sum_{i=1}^{N_m} - \log \frac{\exp\left(z_{i,\mathrm{p}}^+/\tau\right)}{\exp\left(z_{i,\mathrm{p}}^+/\tau\right) + \exp\left(z_{i,\mathrm{p}}^-/\tau\right)} \quad (5)$$

For $z_{i,\mathrm{p}}^+$ and $z_{i,\mathrm{p}}^-$, calculating method is similar, take the former as example, we have the expression Eq. 6, where $M$ represents the number of samples in a batch, $I_{\mathrm{p}(i,j)}$ is an indicator function that equals 1 if sample $j$ has the same label $p$ with sample $i$, and 0 otherwise, $\epsilon$ is a small constant to prevent division by zero:

$$z_{i,\mathrm{p}}^+ = \frac{\sum_{j=1}^M Sim(\Phi(s_i), \Phi(s_j)) \cdot I_{\mathrm{p}(i,j)}}{\sum_{j=1}^M I_{\mathrm{p}(i,j)} + \epsilon} \quad (6)$$

We take contrastive loss in label $p$ as example, label $q$, $r$ and $l$'s loss are consistent with the above equation, they together demonstrate our fine-grained differentiation for AI samples.

The final contrastive learning loss should be the sum of the contrastive loss at different granularities above, so we have Eq. 7, where $l_i$ represents the label $l$ that whether the $i_{th}$ sample belongs to human or AI, $K$ indicates the sum of all samples entering the subdivision module, $\beta$, $\gamma$, $\delta$, $\eta$ are weight coefficients, $\alpha = \beta + \gamma + \delta + \eta$, and $\mathcal{L}_p$, $\mathcal{L}_q$, $\mathcal{L}_r$ and $\mathcal{L}_l$ represents the loss of the subdivision module at different granularities:

$$\mathcal{L}_{contra-tot} = \sum_{i=1}^K \alpha \cdot l_i \cdot \mathcal{L}_{l-human} + (1 - l_i) \cdot (\beta \cdot \mathcal{L}_p + \gamma \cdot \mathcal{L}_q + \delta \cdot \mathcal{L}_r + \eta \cdot \mathcal{L}_{l-AI}). \quad (7)$$

Through multi-grained contrastive loss function propagation, the model can distinguish the differences between AI texts in a fine-grained manner. We introduce the cross-entropy loss function Eq. 1 to drive the model to improve performance in the final binary classification task, the final loss is:

$$\mathcal{L}_{final-loss} = \lambda \cdot \mathcal{L}_{contra-tot} + (1 - \lambda) \cdot \mathcal{L}_{ce}. \quad (8)$$

## 4 EXPERIMENTS

### 4.1 EXPERIMENTAL SETUP

**Datasets.** We use three datasets that are widely used for detector training and detection. Detailed dataset information is in Appendix D. **HC3** (Guo et al., 2023): A high-quality dataset containing QA

question-answer pairs in numerous fields, each question corresponds to at least one human answer and one machine-generated answer, focusing on multiple open-ended questions such as finance and medicine. **SeqXGPT-Bench** (Wang et al., 2023): A benchmark dataset designed specifically for sentence-level AI generated text detection tasks, containing text generated from multiple LLMs (such as GPT-2, GPT-Neo, GPT-J, LLaMa, and GPT-3). **CheckGPT** (Liu et al., 2024c): This dataset contains 900,000 samples, generated by ChatGPT based on prompts, covering diverse fields such as news, reviews, and literatures.

**Evaluating Metrics.**   We use Accuracy (ACC), F1-score (F1), and Recall as the main standard (since our work can be seen as a classification task). We also introduce AUROC, FPR@95%TPR, and AUPR: AUROC measures the model's overall ability to distinguish positives from negatives across all thresholds; FPR@95%TPR (we use F95T for short) reports the false-positive rate incurred when 95% of positives are recalled (the lower the better); AUPR summarizes the precision-recall curve and reflects minority-class performance when positives are rare.

**Baseline Detectors.**   We select the following five representative detectors as baselines and compare them with HiDet. **SimpleAI** (Guo et al., 2023): Fine-tune the pre-trained RoBERTa model with high-quality datasets. **Watermark** (Kirchenbauer et al., 2023): watermark methods embed the signal during generation, and determine whether the text is generated by AI through detecting the signal. **CoCo** (Liu et al., 2023): By constructing a coherence graph to capture the entity interaction structure of the text and introducing a supervised contrastive learning framework, the model's understanding of language patterns is enhanced. **RADAR** (Hu et al., 2023): Using the adversarial learning framework of GAN, model shows excellent robustness and transferability. **PECOLA** (Liu et al., 2024b): The noise introduced by random perturbations is reduced through selective perturbation strategies, and contrastive learning strategy is further used to enhance robustness. We also compare with DeTeCtive (Guo et al., 2024), Binoculars (Hans et al., 2024), Fast-detectgpt (Bao et al., 2023), Dna-gpt (Yang et al., 2023), detailed in Appendix J.

### 4.2 RESULTS AND ANALYSIS

**Task 1: Detecting human texts and AI texts.**   Results are shown in the Task 1 column of Table 1. For HC3 and SeqXGPT datasets, our method outperforms all baseline detectors in all evaluating metrics. For the CheckGPT dataset, we achieve the best in F1, ACC, and also achieve the second best in overall recall. Further, using the comprehensive evaluating metric of F1 to illustrate, HiDet is 0.72% higher than the second place on the SeqXGPT dataset and 4.11% higher than the second place on the CheckGPT dataset. For the HC3 dataset, all baseline detectors perform well, HiDet still achieves a certain breakthrough with F1 0.54% higher than the second place. The above results show that the cross-data adaptability of our two-module framework is commendable, and acquires the most advanced performance in Task 1.

**Task 2: Detecting human texts, AI texts and humanized AI texts.**   We humanize the AI texts in the HC3, SeqXGPT and CheckGPT datasets, and retrain the baseline detector based on the humanized datasets to compare with HiDet. Results correspond to the Task 2 column in Table 1. Our detector HiDet has achieved SOTA performance on the three humanized datasets. In terms of recall, F1, and ACC, HiDet is 17.06%, 0.90%, and 3.08% higher than the second place on the HC3 dataset, and 3.58%, 0.96%, and 1.05% higher than the second place on the SeqXGPT dataset. On the CheckGPT dataset, recall and ACC are both the first place, and F1 reaches the runner-up performance. Furthermore, for the detector CoCo, which also has outstanding performance, although its performance on the CheckGPT dataset is comparable to ours, it relies on the extraction of entities in texts and the construction of a coherence graph. If the AI texts are relatively concise and short, its performance will suddenly drop because of the failure in building coherence graphs.

We also notice that some detectors performed better on Task 2 than on Task 1, this suggests the success of robust strategies applied by these detectors. These relatively high-performing detectors are specifically trained for robustness against humanizing methods. Some humanized texts may introduce artifacts due to its imitative nature, and these detectors have enhanced their recognition capabilities through adversarial training.

| Dataset | Detectors | Task 1 | | | Task 2 | | |
|---|---|---|---|---|---|---|---|
| | | Recall | F1 | ACC | Recall | F1 | ACC |
| HC3 | SimpleAI | 94.32 | 94.31 | 94.32 | 71.58 | 79.25 | 96.44 |
| | Watermark | 94.75 | 95.13 | 94.88 | 76.03 | 69.05 | 55.16 |
| | CoCo | 99.31 | 98.30 | 98.42 | 58.18 | 95.09 | 94.44 |
| | RADAR | 89.57 | 90.39 | 89.57 | 79.20 | 89.40 | 81.75 |
| | PECOLA | 99.25 | 99.24 | 99.23 | 64.03 | 70.59 | 95.44 |
| | **HiDet** | **99.78** | **99.78** | **99.80** | **96.26** | **95.99** | **99.52** |
| SeqXGPT | SimpleAI | 94.38 | 94.36 | 94.37 | 81.74 | 86.18 | 97.04 |
| | Watermark | 96.30 | 95.92 | 96.07 | 76.22 | 68.80 | 54.61 |
| | CoCo | 82.36 | 79.54 | 80.67 | 93.65 | 93.06 | 98.16 |
| | RADAR | 61.15 | 54.16 | 61.37 | 66.77 | 72.08 | 58.51 |
| | PECOLA | 90.83 | 90.82 | 90.82 | 74.98 | 80.30 | 96.06 |
| | **HiDet** | **96.70** | **96.64** | **96.66** | **98.92** | **97.13** | **98.97** |
| CheckGPT | SimpleAI | 87.82 | 88.78 | 88.77 | 88.58 | 70.21 | 90.52 |
| | Watermark | **97.06** | 72.26 | 75.69 | 76.54 | 69.56 | 56.22 |
| | CoCo | 84.90 | 85.97 | 84.55 | 85.63 | **98.21** | 96.60 |
| | RADAR | 63.26 | 63.01 | 63.04 | 72.71 | 75.35 | 61.94 |
| | PECOLA | 87.16 | 86.79 | 86.82 | 69.14 | 75.63 | 96.90 |
| | **HiDet** | 93.33 | **92.89** | **93.55** | **96.08** | 95.71 | **99.63** |

Table 1: Combined performance results across original and humanized datasets. The best number is highlighted in **bold**, while the second best one is underlined, below are the same.

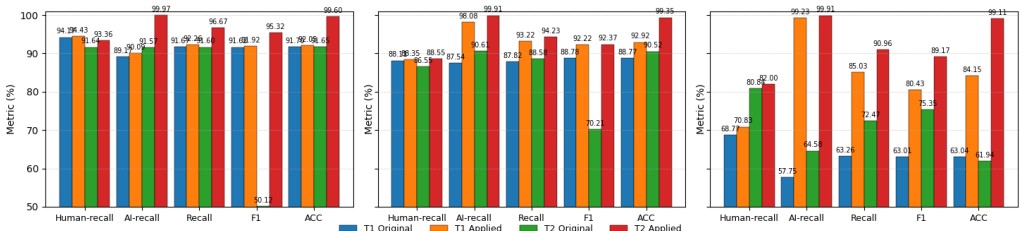

Figure 3: "Patching" results for fine-tuned DeBERTa, SimpleAI, and RADAR.

**"Patching" results for existing detectors.** We further explore the effect of our framework in "patching" existing detectors, given that the coarse module plays an important role in the overall detection effect, its initial screening of data directly affects the training and detection of the subdivision module. We replace the first module based on traditional classification loss with fine-tuned DeBERTa (Wang et al., 2024), along with baseline detectors SimpleAI (Guo et al., 2023) and RADAR (Hu et al., 2023). Results are shown in Figure 3. Taking the F1 score as an example, for Task 1, the two detectors still have steady improvements, SimpleAI from 88.78% to 92.22%, an increase of 3.44%; RADAR from 63.01% to 80.43%, an increase of 17.42%. Furthermore, when facing humanized data sets, the performance improvement is particularly obvious, such as SimpleAI's F1 from 70.21% to 92.37%, a huge leap of 22.16%. This further proves one of our intention of using two-module detection: To give the existing detector robustness when facing large-scale humanized AI texts, offering a second checkpoint to capture those humanized AI texts that bypass detection.

**Deeper exploration for coarse module.** For coarse module's threshold, it is determined by experimental parameter adjustment. Another advantage of Hidet is that it is not sensitive to this threshold

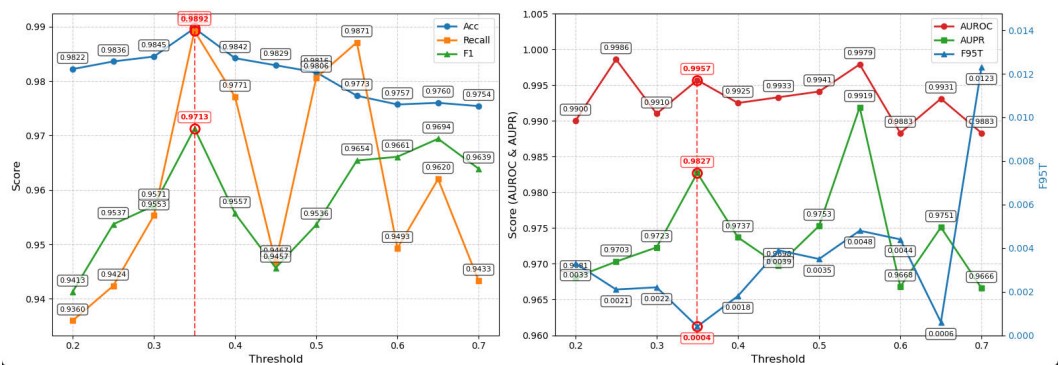

Figure 4: Performance under different thresholds.

(the second module provides a backup, machine text that the first module misses will still be detected by the second module). We chose 0.35 as threshold, the results are shown in Figure 4.

### 4.3 PERFORMANCE FOR UNSEEN ATTACKS

To further demonstrate that Hidet's multi-stage and multi-granularity strategy can effectively capture the common features of AI humanized texts, we design several experiments to demonstrate Hidet's resilience to attacks:

**Randomly masked humanizing methods.** For the first experiment: during training, we randomly mask several humanizing methods (removing spaces, simulating spelling errors, and repeating sentences) and construct three new test datasets. Each dataset consisted of human texts, machine texts, and machine texts humanized using one of the masked strategies (a 1:1:1 split). We evaluat the newly trained Hidet on these new datasets, with the results shown in Table 2.(The first three rows in the table show the performance of the newly trained Hidet under three unseen attacks. The last row is the performance of Hidet on the seqxgpt dataset in the paper. We present it here for better comparison.) Conclusions can be draw that although Hidet does experience a performance degradation of 1-2%, it does not collapse and still demonstrate excellent performance.

**Newly introduced humanizing methods.** In order to simulate the generalization of Hidet in the face of new humanizing strategies in reality, we introduce three unseen attack methods from Wang's work (Wang et al., 2024), they are: Homoglyph Alteration (change English characters into visually similar Unicodes), Format Character Editing (change or insert formatting characters, including zero-width whitespace insertion, and shift character editing), Emoji Co-Generation (compulsorily generate or insert an emoji after finishing each sentence while recurrent generation and remove all the emojis after finishing the whole text). We also construct a new test dataset (method is the same as above) and test the generalization of Hidet under these new attacks, results shown in Table 2: Even in the face of new humanized methods, Hidet performs very well, which demonstrates its robustness against new attacks and indicates that it has successfully learned the commonalities of humanizing strategies through our designed strategies.

**Hybrid humanizing attacks.** We consider that in real life, attackers may not be limited to using only one humanizing method, we select representative humanizing methods from characters, words, sentences, and paragraphs and mix them to simulate this situation and test Hidet, results shown in Table 2. As can be seen from the table, Hidet has stable performance and maintains good robustness in the face of hybrid attacks.

### 4.4 ABLATION STUDIES

To validate the effectiveness of our framework, we conduct ablation studies on both module design and contrastive learning strategy. First, we verify that both the coarse and subdivision modules are

| Attack Type | Acc | Recall | F1 | AUROC | F95T | AUPR |
|---|---|---|---|---|---|---|
| **Masked Attacks** | | | | | | |
| **spaceremove** | 0.9640 | 0.9558 | 0.9452 | 0.9915 | 0.0230 | 0.9817 |
| **typos** | 0.9634 | 0.9594 | 0.9562 | 0.9939 | 0.0135 | **0.9915** |
| **repeatsent** | 0.9709 | 0.9610 | 0.9552 | **0.9957** | 0.0103 | 0.9910 |
| **Hidet-sota** | **0.9897** | **0.9892** | **0.9713** | **0.9957** | **0.0004** | 0.9827 |
| **Unseen Attacks** | | | | | | |
| **Homoglyph Alteration** | 0.8783 | 0.8938 | 0.8374 | 0.9507 | 0.1675 | 0.9069 |
| **Format Character Edit** | 0.9750 | 0.9400 | 0.9616 | 0.9932 | 0.0100 | 0.9893 |
| **Emoji Co-Generation** | 0.9733 | 0.9650 | 0.9592 | 0.9955 | 0.0125 | **0.9912** |
| **Hidet-sota** | **0.9897** | **0.9892** | **0.9713** | **0.9957** | **0.0004** | 0.9827 |
| **Hybrid Attacks** | | | | | | |
| **punctremove+reverse** | 0.9853 | 0.9803 | 0.9778 | 0.9963 | 0.0030 | 0.9937 |
| **backtrans+spaceremove** | 0.9864 | 0.9816 | 0.9794 | 0.9956 | 0.0020 | 0.9933 |
| **para+typos** | 0.9864 | 0.9817 | **0.9793** | **0.9978** | 0.0021 | **0.9961** |
| **para+reverse** | 0.9860 | 0.9813 | 0.9788 | 0.9964 | 0.0023 | 0.9939 |
| **Hidet-sota** | **0.9897** | **0.9892** | 0.9713 | 0.9957 | **0.0004** | 0.9827 |

Table 2: Comparative Performance on different attack types, F95T means FPR@95%TPR.

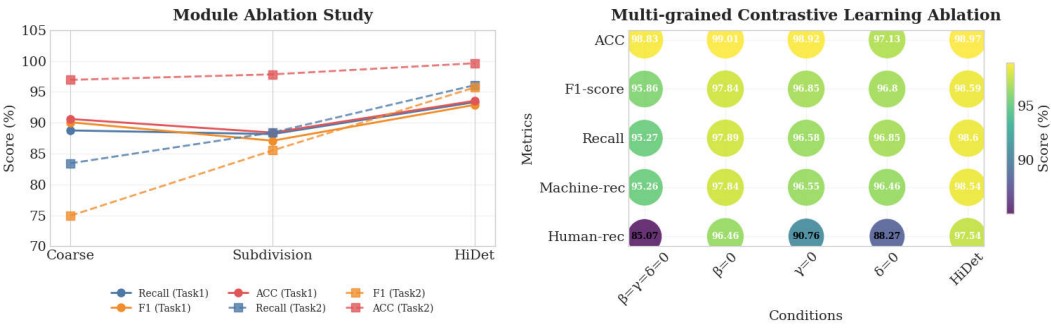

Figure 5: Ablation studies for module design and contrastive learning strategy.

indispensable by training them separately on two tasks: detecting original AI texts (Task 1) and humanized AI texts (Task 2). Results show that while the coarse module performs better on simple detection tasks, the subdivision module excels at detecting humanized texts (F1 increased by 10% in Task 2), with HiDet achieving the best performance in both tasks by leveraging their complementary strengths. Second, we evaluate our multi-grained contrastive learning approach on the subdivision module, demonstrating significant improvements over baselines (15% increase in human recall and 10% in F1), proving its effectiveness in capturing subtle patterns that prevent humanized AI texts from evading detection, detailed results in Appendix I.

## 5 CONCLUSION

In this paper, we propose a coarse-to-fine AI generated text detector model and a novel training paradigm. The coarse module quickly screens the original AI texts, and the subdivision module uses multi-grained contrastive learning to carefully distinguish different levels of humanized AI texts. Our detector HiDet has achieved SOTA performance in two main tasks. By decoupling the complete process into simple samples detection and difficult samples detection, we set up solid defence towards AI humanizing attacks. Moreover, our subdivision module is a plug-and-play patch that can be easily applied to existing detectors to further improve their performance. We hope HiDet can better assist AI text detection in real life, and that this hierarchical and multi-grained contrastive learning framework can bring new maps and ideas to researchers in this field.

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

## A  BROADER IMPACTS

The rapid development of LLMs has enabled a large amount of AI-generated texts to be obtained quickly and at low cost. Given that it may lead to academic fraud, phishing emails, the spread of false information and other problems, detecting and monitoring AI-generated texts is undoubtedly a top priority. However, due to the fragility of current AI content detectors and the diversity of text humanizing methods, AI texts can easily bypass detection after humanizing. Therefore, the development of robust AI content detectors is urgent. Our paper introduces a new robust AI content detector training paradigm, which demonstrates SOTA performance in multiple benchmarks. These advances will bring the green development and use of LLMs with new power. In addition, our subdivision module can be used as a "patch" to further improve the performance of current detectors when facing large-scale humanized AI texts, which shows that our method has broad prospects for practical application and rich significance.

## B  LIMITATIONS AND FUTURE WORK

In this paper, we take into consideration that AI texts may use different humanizing methods to evade the detector and thus use multi-grained contrastive learning to strengthen the detector in a targeted manner. However, since we mainly focus on the robustness task of detection, Hidet's generalization performance on other datasets may not be as good as zero-shot works. In the future, one possible consideration in the field of AI text detection is how to combine the zero-shot method based on statistics with the supervised training method based on deep learning to achieve better robustness and generalization. We will continue to work in this direction.

## C  DETAILED HUMANIZING METHODS AND ITS LEVELS

Following Zhou's work (Zhou et al., 2024), four types of humanizing methods are classified below:

1. **char level**: Attacks at this level include space deletion, space addition (Cai and Cui, 2023), capitalization typo simulation, punctuation deletion, and random word merging.

2. **word level**: Attacks at this level include keyboard spelling errors, which replaces characters in similar keyboard positions; swaping adjacent characters, inserting irrelevant characters, and deleting specific characters, which simulate human negligence when typing; word spelling errors, which simulates users' incorrect spelling of words through a predefined spelling error dictionary; adverb perturbations, which randomly inserts relevant adverbs before verbs in the original text; word replacement, which uses the BERT model (Devlin et al., 2019) to replace words in the text with synonyms.

3. **sentence level**: Attacks at this level include adding irrelevant sentences; repeating parts of sentences; randomly selecting sentences for back-translation; and sentence-level replacement, which randomly masks 2 to 5 sentences in the original text and replaces them using the BART-large model (Lewis et al., 2020).

4. **paragraph level**: Attacks at this level include rewriting using the Dipper interpreter (Krishna et al., 2024); back-translation using the Helsinki-NLP model (Tiedemann and Thottingal, 2020); and rearrangement of paragraph structure.

Given original AI texts $X$, we thus have humanized AI texts set:

$$X_{humanized} = \{X_{char}, X_{word}, X_{sent}, X_{para}\}$$

which is illustrated in section 3.1.

## D  DETAILED CONSTRUCTION OF DATASET

For the simple binary classification task of detecting human texts and original AI texts, the distribution of our samples is shown in Table 3. We ensure the distribution of human texts and AI texts (machine texts) is approximately 1:1. The two-tuple (human, machine) in the table represents the number of human texts and the number of AI texts. For the watermark detector, since it focuses on the hidden

| Dataset | Train | Test | Valid |
|---------|-------|------|-------|
| CheckGPT | (2000,2000) | (1921,2078) | (2500,2500) |
| HC3 | (5040,5000) | (5040,5000) | (2000,2000) |
| SeqXGPT | (2467,2533) | (1928,1872) | (1005,995) |

Table 3: Detailed composition of the dataset for detecting human texts and original AI texts.

| Watermark | Human | Machine |
|-----------|-------|---------|
| CheckGPT | 566 | 570 |
| HC3 | 438 | 538 |
| SeqXGPT | 600 | 520 |

Table 4: Detailed composition of the dataset for watermarks.

| Dataset | Train | Test | Valid |
|---------|-------|------|-------|
| CheckGPT | (500,8500) | (1101,23887) | (500,8490) |
| HC3 | (500,7500) | (1500,22500) | (500,7500) |
| SeqXGPT | (500,7500) | (1500,21004) | (500,6494) |

Table 5: Detailed composition of the dataset for detecting human texts and humanized AI texts.

| Watermark | Human | Machine |
|-----------|-------|---------|
| CheckGPT | 566 | 8550 |
| HC3 | 438 | 8098 |
| SeqXGPT | 600 | 520 |

Table 6: Detailed composition of the humanized dataset for watermarks.

singal embedded in the data and does not require training, it only needs to build a test set. While it takes a long time to process the watermark on the dataset, we did not generate a large test set. The data are shown in Table 4. For the more realistic task of detecting human texts and humanized AI texts, the distribution of our samples is shown in Table 5. We select 500 human texts and 500 original AI texts from the dataset respectively, and perform 16 humanizing methods on the AI texts at the character, word, sentence, and paragraph levels, thereby constructing a perturbation dataset containing human texts, original AI texts, and humanized AI texts. Therefore, the perturbed datasets are mostly composed of AI texts, which are consistent with the current trend of a variety of machine text humanizing methods and a flood of generation sources. For the watermark detector, similarly, after the AI texts are injected with the watermark, we humanize them in sixteen different ways and explore whether these humanizing attacks will cause the watermark to be covered and invalid. The data distribution is shown in Table 6.

## E    COMPARISON OF DETECTORS IN COSTS

| Detectors | Preprocess Time | Preprocess Memory | Train Time | Train Memory |
|-----------|-----------------|-------------------|------------|--------------|
| PECOLA | 22:17:04 | 642 | 00:08:55 | 9228 |
| CoCo | 04:42:56 | 10520 | 00:33:27 | 14556 |
| Watermark | 153:21:22 | 1290 | 00:01:31 | 840 |
| Contra | no need | no need | 00:12:40 | 10840 |
| Coarse | no need | no need | 00:08:40 | 10840 |
| Subdivision | no need | no need | 00:01:36 | 10840 |

Table 7: Detailed comparison of the detectors in terms of costs. Preprocess time means data should be preprocessed before training.

The baseline detectors SimpleAI and RADAR are similar to coarse module (which are also the subjects of our patching experiments) thus are not considered here. Shown in Table 7, Contra means we directly use contrastive learning to train the detector. The data processing and training time are formed in hour: minute :second, and the memory occupied is in MB. Watermark does not require additional training of the detector, and its training time is temporarily expressed as the detection time. We uniformly set the training batch size to 16, the training round to 15, and trained on NVIDIA RTX A6000. The high cost of contrastive learning is that it needs to calculate the similarity between all

sample pairs, which has a time complexity of $O(n^2)$ while cross entropy of $O(n)$, which will mainly affect training time rather than memory usage. Compared with using contrastive learning directly, the subdivision module only needs to process the filtered samples, and the training time is reduced from 12 minutes to 1 minute 36 seconds. In addition, although PECOLA, CoCo and Watermark have achieved good performance in detecting AI texts, their cumbersome and lengthy data preprocessing process deserves high attention and needs to be considered seriously. Taking CoCo as an example, it is slow in extracting entity graphs from texts, takeing nearly 5 hours to extract 8,000 training samples. When large-scale machine text needs to be detected in real life, such as the 23005 samples in our test data set, its data preprocessing takes an astonishing 17 hours, shown in Table 9. These situations remind us that if we want to make an AI content detector with advanced performance and practical significance, the time issue of data processing needs to be paid great attention to.

| PECOLA | Augment | Select | Total |
|---|---|---|---|
| Train | 00:03:44 | 22:13:20 | 22:17:04 |
| Eval | 00:03:16 | 19:26:45 | 19:30:01 |
| Test | 00:09:48 | 56:56:28 | 57:06:16 |

| CoCo | Extract | Build | Total |
|---|---|---|---|
| Train | 04:41:04 | 00:00:31 | 04:42:56 |
| Eval | 02:03:20 | 00:00:23 | 02:03:43 |
| Test | 17:06:38 | 00:01:21 | 17:07:59 |

Table 8: Detailed time cost for PECOLA in building train, eval and test sets. Augment means data augmentation, Select means its selecting strategy.

Table 9: Detailed time cost for CoCo in building train, eval and test sets. Extract means extracting entity, Build means building graphs according to the extracted entities.

## F  AI-GENERATED TEXT DETECTORS

In order to prevent AI-generated texts from being abused, numbers of detectors have been proposed by researchers, thus consolidating the defense line of text detection. We classify the existing detectors into the following four categories:

**Statistical and mathematical based detectors**: Using information entropy, cross perplexity, word frequency statistics and other features to perform zero-shot detection. Mithcell (Mitchell et al., 2023) quantifies the difference between machines and human in word selection via conditional probability curvature. Su (Su et al., 2023) applies log-rank information to detect. Open source detecting platform GPTZero and GLTR (Tian and Cui, 2023; Gehrmann et al., 2019) are also included.

**Watermark based detectors**: Watermark detection algorithms in AI texts detection track the source of generated texts by embedding invisible identifiers. Representative works include: (Gu et al., 2022; Liu et al., 2024a; Hou et al., 2024; Lu et al., 2024). Among them, Kirchenbauer (Kirchenbauer et al., 2023) adds a fixed weight to the logit value of the predefined "green word list" and determine whether the text is generated by the model by counting the proportion of green words in the texts.

**Classifier based detectors**: Researchers (Chen et al., 2023; Miao et al., 2024; Mireshghallah et al., 2024; Wang et al., 2023; Liu et al., 2024c) typically employ RoBERTa (Liu et al., 2019) as the backbone architecture for training supervised binary classifiers. Notable developments are seen in OpenAI's official detection toolkit (Solaiman et al., 2019a) and RADAR (Hu et al., 2023), which enhances adversarial robustness against perturbation attacks through paraphrase-based adversarial training.

**Other methods based detecors**: Soto (Soto et al., 2024b) uses style representations, Huang (Huang et al., 2024) takes advantage of siamese neural network, Krishna (Krishna et al., 2024) achieves success through retrieval methods. Zhu (Zhu et al., 2023) innovatively queries LLM to detect LLM-generated texts.

## G  TRAINING DETAILS

In the experiment,we use SimCSE-RoBERTa(Gao et al., 2021) as encoder and freeze its embedding module. For coarse module, we employ AdamW(Loshchilov and Hutter, 2017), learing rate at

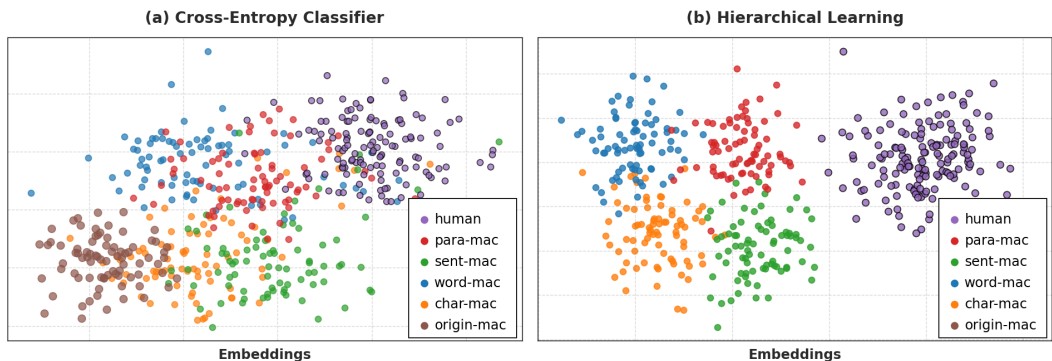

Figure 6: Illustration of contrastive learning and cross entropy on final texts' embeddings. Para-mac reffers to text using humanizing method of paragraph level, rest are the same.

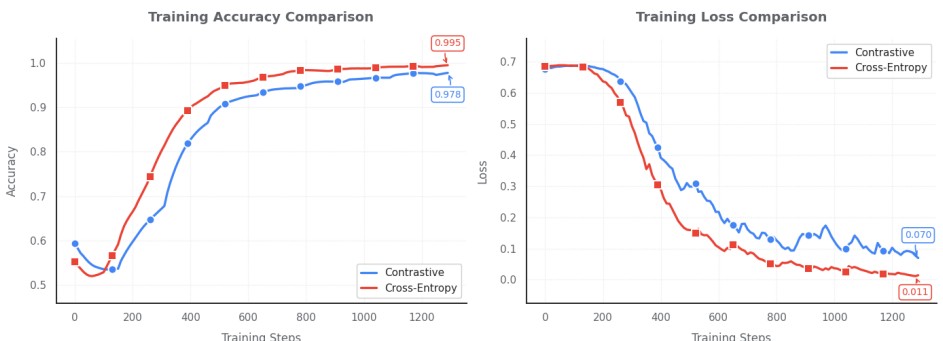

Figure 7: Contrastive loss VS cross-entropy loss on HC3.

2e-5, $\beta_1$ at 0.9, $\beta_2$ at 0.98, warmed up 2000 steps, weight decay at 1e-4, maximum input token length is 512, threshold 0.35. For subdivision module, we set learing rate at 1e-5, warmup at 1000, $\beta = \gamma = \delta = \eta = 1$ and epochs 15. For baselines, we retrain them on the three datasets of HC3, SeqXGPT and CheckGPT according to the method described in their papers. RADAR does not provide source code, so we use the pretrained model they provide. In addition, the watermark-based detector does not need to be trained, its processing work is to add watermarks to the input dataset.

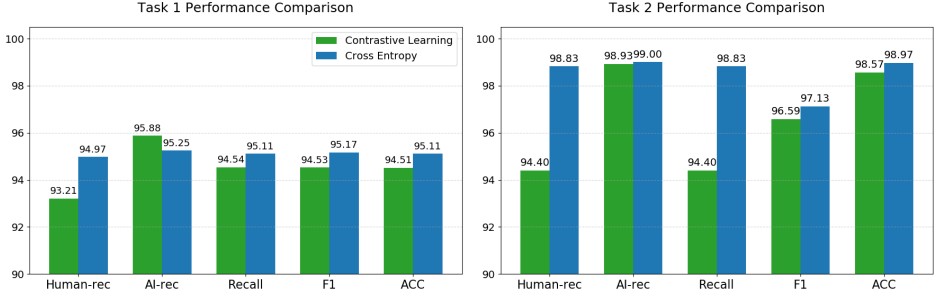

Figure 8: Comparison of coarse module with contrastive learning and cross-entropy on SeqXGPT.

## H  CONTRASTIVE LEARING IN DETECTORS

There have been works showing that contrastive learning has excellent performance in the field of natural language processing (Cheng et al., 2023). MixCSE (Zhang et al., 2022b), SimCSE (Gao et al., 2021), VaSCL (Zhang et al., 2022a) use unsupervised contrastive learning framework to enhance the semantic discrimination ability of the model; CoCo (Liu et al., 2023) use supervised contrastive learning to make the model pay more attention to difficult negative samples in low-resource scenarios; Soto and Guo (Soto et al., 2024a; Guo et al., 2024) use contrastive learning to distinguish the style features of human and machine writing. By narrowing the distance between positive samples and increasing the distance between negative samples, contrastive learning has shown great potential in training AI content detectors (Liu et al., 2024b).

We introduce contrastive learning strategy in the subdivision module. For the coarse screening module, only the cross entropy loss function is used. This has several advantages: first is better performance, shown in Figure 8. Compared with coarse module using contrastive learning, Hidet reaches the lead in AI recall, average recall, ACC and F1. In addition, HiDet using contrastive learning in both stages also shows performance degradation, as the human recall rate regressed from 98.83% to 94.40%. Second, coarse module based on cross entropy converges faster and more stably, and its training accuracy is higher than that of the module based on contrastive learning, illustrated in Figure 7.

## I  DETAILED ABLATION STUDIES

**Ablation study for modules.**   In this experiment, we aim to verify that the coarse module and subdivision module are indispensable. We directly train the two modules on the CheckGPT dataset in Task 1 and humanized CheckGPT dataset in Task 2, results are shown in Table 10. Since the subdivision module is designed for detecting AI texts humanized in different level, it performs worse than the coarse module in Task 1, which is a simple task of detecting original AI texts, but significantly outperforms the coarse module in Task 2, a task which involves disguised AI texts (F1 increased by nearly 10% from 74.95% to 85.51%). At the same time, all metrics of HiDet in the two tasks are better than the previous two. This setting of making the coarse module focus on filtering simple samples and the subdivision module focus on detecting difficult samples enables the two modules to perform their respective duties, achieving the results of "one plus one greater than two".

| Module | Task 1 | | | Task 2 | | |
|---|---|---|---|---|---|---|
| | Recall | F1 | ACC | Recall | F1 | ACC |
| Coarse | 88.75 | 90.07 | 90.60 | 83.43 | 74.95 | 96.94 |
| Subdivision | 88.16 | 87.09 | 88.40 | 88.42 | 85.51 | 97.83 |
| HiDet | **93.33** | **92.89** | **93.55** | **96.08** | **95.71** | **99.63** |

Table 10: Results of ablation study for modules. We directly train and apply each module on the two tasks.

**Ablation study for multi-grained contrastive learning.**   In this experiment, we aim to verify the effectiveness of multi-grained contrastive learning on the subdivision module. While keeping the coarse module the same, we retrain the subdivision module of different granularity on SeqXGPT humanized dataset and set it as the contrast, results are shown in Table 11. The subdivision module using multi-grained contrastive learning is better than the former in all metrics. In comparison with the basical one, a significant increase of nearly 15% in human recall and a notable improvement of 10% in F1 are witnessed. The result proves that the multi-grained contrastive learning we designed promotes better understanding of details and effectively solve the current problem of AI texts escaping detection through humanizing methods.

| Weight Coefficient | Metrics | | | | |
|---|---|---|---|---|---|
| | Human-rec | Machine-rec | Recall | F1-score | ACC |
| $\beta = \gamma = \delta = 0$ | 85.07 | 96.46 | 90.76 | 88.27 | 97.54 |
| $\beta = 0$ | 95.26 | 97.84 | 96.55 | 96.46 | 98.54 |
| $\gamma = 0$ | 95.27 | 97.89 | 96.58 | 96.85 | 98.60 |
| $\delta = 0$ | 95.86 | 97.84 | 96.85 | 96.80 | 98.59 |
| HiDet | **98.83** | **99.01** | **98.92** | **97.13** | **98.97** |

Table 11: Results of ablation study for multi-grained contrastive learning. We drop each component of the loss function 7 and retrain the subdivision module.

## J  SUPPLEMENTARY EXPERIMENTS

To better evaluate our work ,we introduce new baselines DeTeCtive (Guo et al., 2024), Binoculars (Hans et al., 2024), Fast-detectgpt (Bao et al., 2023), Dna-gpt (Yang et al., 2023) and introduce more evaluation indicators. The results are shown in Table below, we do this on SeqXGPT dataset.

On both tasks, Hidet still achieved sota performance. This demonstrates the necessity of Hidet's new paradigm of multi-stage detection and the effectiveness of multi-granularity contrastive learning in distinguishing humanized texts.

Table 12: Task 1: Detecting human vs. machine texts.

| Detectors | ACC | Recall | F1 | AUROC | FPR@95%TPR | AUPR |
|---|---|---|---|---|---|---|
| Fast-detect | 0.6460 | 0.5960 | 0.6274 | 0.6697 | 0.8420 | 0.6183 |
| Binoculars | 0.8070 | 0.8160 | 0.8087 | 0.8630 | 0.7200 | 0.8771 |
| Dna-gpt | 0.7150 | 0.8120 | 0.7273 | 0.7100 | 0.5950 | 0.7223 |
| DeTeCtive | 0.9461 | 0.9186 | 0.9453 | 0.9876 | 0.0748 | 0.9887 |
| SimpleAI | 0.9437 | 0.9438 | 0.9436 | 0.9831 | 0.0768 | 0.9858 |
| Watermark | 0.9607 | 0.9630 | 0.9592 | **0.9990** | **0.0000** | **0.9990** |
| CoCo | 0.8067 | 0.8236 | 0.7954 | 0.8942 | 0.2326 | 0.8978 |
| RADAR | 0.6137 | 0.6115 | 0.5415 | 0.6737 | 0.7230 | 0.6238 |
| PECOLA | 0.9082 | 0.9083 | 0.9082 | 0.9705 | 0.1680 | 0.9735 |
| **HiDet** | **0.9666** | **0.9670** | **0.9664** | 0.9931 | 0.0208 | 0.9923 |

Table 13: Task 2: Detecting human, machine, and humanized machine texts.

| Detectors | ACC | Recall | F1 | AUROC | FPR@95%TPR | AUPR |
|---|---|---|---|---|---|---|
| Fast-detect | 0.4159 | 0.3913 | 0.5569 | 0.5781 | 0.9950 | 0.9516 |
| Binoculars | 0.6758 | 0.6706 | 0.7951 | 0.7528 | 0.8996 | 0.9775 |
| Dna-gpt | 0.1979 | 0.1455 | 0.2486 | 0.4929 | 0.9890 | 0.9469 |
| DeTeCtive | 0.8502 | 0.8824 | 0.4446 | 0.9262 | 0.2128 | 0.3528 |
| SimpleAI | 0.9704 | 0.8174 | 0.8618 | 0.9550 | 0.2233 | 0.9962 |
| Watermark | 0.5461 | 0.7620 | 0.6879 | **0.9990** | **0.0000** | **0.9999** |
| CoCo | 0.9816 | 0.9365 | 0.9306 | 0.9732 | 0.0054 | 0.9789 |
| RADAR | 0.5850 | 0.6677 | 0.7207 | 0.7382 | 0.6726 | 0.9723 |
| PECOLA | 0.9606 | 0.7498 | 0.8030 | 0.9410 | 0.9952 | 0.2673 |
| HiDet | **0.9897** | **0.9892** | **0.9713** | 0.9957 | 0.0004 | 0.9827 |

## K  THE USE OF LARGE LANGUAGE MODELS (LLMS)

Our work uses LLM as a general auxiliary tool. Its functions include assisting with image rendering and beautification, assisting with paper writing and polishing, and data screening during training.

