# OpenReview forum: "HiDet: Hierarchical AI Text Detection via Coarse Filtering and Multi-Grained Contrastive Learning"
_ICLR.cc/2026/Conference — ICLR 2026 Conference Withdrawn Submission_

### Official Review · Reviewer_96t2 · 2025-10-31

**Soundness:** 2
**Presentation:** 2
**Contribution:** 2
**Rating:** 4
**Confidence:** 3

**Summary:**

The author proposes the HiDet hierarchical detection framework: a coarse module (traditional cross-entropy classifier) ​​to a fine-module (multi-granularity contrastive learning) for AI generated text detection.
While the subdivision module can be embedded as independent plugins into existing detectors, significantly improving their robustness to humanized text.

**Strengths:**

1. In the Plug-and-Play experiment, multiple baseline methods improve more than 15% on humanized data, directly verifying that "subdivision module can be easily patched to existing detectors."

2. HiDet shows a performance decrease of less than 3% on F1 score against most unseen attacks and mixed attacks, indicating that multi-granularity contrastive learning learns not just surface perturbations, but deeper common AI characteristics.

3. Ablation study shows that removing any granularity of contrastive loss reduced human recall by 3%, proving that each granularity contributes to effectiveness.

**Weaknesses:**

1.The experimental results in Table 1 confirm that the HiDet method achieves SOTA performance on most scenarios and datasets, but it doesn't completely outperform the second-best method. The author could supplement this with results on more challenging datasets. For example, Table 2 should include performance comparisons to other detector methods.

2.The author emphasizes that some high-performing detectors may exhibit length sensitivity, performing worse on shorter texts. However, the author does not conduct experiments dividing texts by length to support their point.

3.The author demonstrates the efficiency of their method during the training phase in Table 7, but they do not mention the inference cost, which is more worthy of consideration in real-world deployment.

Typos
Section4.3.1 evaluat - evaluate

**Questions:**

1.Could you explain the implementation details of HiDet at the subdivision stage at inference time? Are you following DeTeCtive's work by using the KNN algorithm to classify queries against existing databases?

2.Could you please explain the results in Tables 12 and 13, which show that Watermark outperforms the HiDet method on metrics such as AUROC?

---

### Official Review · Reviewer_ACSb · 2025-11-01

**Soundness:** 4
**Presentation:** 3
**Contribution:** 2
**Rating:** 4
**Confidence:** 4

**Summary:**

The paper presents HiDet, a hierarchical AI text detection framework that aims to address the growing challenges posed by humanized AI-generated texts. HiDet is designed to improve the detection accuracy of both original AI texts and humanized versions, which have become increasingly difficult to identify due to various text manipulation techniques. The framework is composed of two primary modules: a coarse module that filters out easily detectable AI texts and a subdivision module that applies multi-grained contrastive learning to more accurately detect humanized AI texts. The paper argues that traditional AI text detectors are vulnerable to humanizing attacks and that HiDet's multi-stage detection process provides a robust defense against these sophisticated evasion techniques.

**Strengths:**

**Hierarchical Framework:** The hierarchical structure of HiDet, with its coarse and subdivision modules, ensures that both simple and difficult cases are handled appropriately. This allows for more efficient and accurate detection across various scenarios.

**Multi-grained Contrastive Learning:** The introduction of multi-grained contrastive learning in the subdivision module enhances the model's ability to distinguish between AI-generated texts and humanized versions. This adds robustness to HiDet, making it highly effective against different levels of humanizing methods.

**State-of-the-Art Performance:** HiDet achieves SOTA performance in detecting both original and humanized AI texts, demonstrating its capability to handle a wide range of text manipulations.

**Practical Applications:** The subdivision module can be added to existing detectors as a plug-and-play solution, offering a practical way to improve detection performance without needing to rebuild entire systems.

**Weaknesses:**

**Limited Baseline Comparison:** While the paper demonstrates HiDet's performance against some baseline detectors, the selection of baselines is somewhat limited. More comparisons with a broader set of state-of-the-art detectors and real-world models would strengthen the evaluation:

- DetectLLM: Leveraging Log Rank Information for Zero-Shot Detection of Machine-Generated Text
- IPAD: Inverse Prompt for AI Detection - A Robust and Interpretable LLM-Generated Text Detector
- Spotting LLMs With Binoculars: Zero-Shot Detection of Machine-Generated Text


**Humanized Methods Exploration:** The paper does a great job outlining different humanizing techniques, but further exploration into additional or emerging methods of text manipulation (such as newer forms of back-translation or syntax-based perturbations) could enhance its robustness even more.

**Dataset Scope:** Although the paper uses well-known datasets (HC3, CheckGPT, SeqXGPT), the datasets used for evaluating humanized AI texts are relatively narrow. More diverse datasets that include a wider variety of languages, domains, and humanizing methods would provide better insight into the generalization ability of HiDet:

- DetectRL: Benchmarking LLM-Generated Text Detection in Real-World Scenarios

**Questions:**

**Adaptability:** How well does HiDet adapt to new, unseen humanizing techniques not covered in the training datasets? Would the system require frequent retraining or could it be made more adaptable to these changes in the text generation landscape?

**Computational Cost:** Is the computational cost of HiDet a significant concern, especially in real-time or large-scale applications, given its two-stage detection process and multi-grained contrastive learning?

**Generalizability:** Are the defined humanizing methods in HiDet sufficient to generalize across all potential future text manipulation techniques, especially as AI-generated texts evolve with new, more sophisticated humanizing methods?

---

### Official Review · Reviewer_BjXp · 2025-11-04

**Soundness:** 1
**Presentation:** 1
**Contribution:** 3
**Rating:** 2
**Confidence:** 4

**Summary:**

This paper proposes a new humanized AI text detection model, HiDet. HiDet uses two phases for detecting AI-generated texts and humanized version of AI-generated text. At the coarse type, it discriminates certainly AI-generated texts from possibly human-written or humanized texts, using a learned classifier. In the finer type, it further discriminates humanized texts from human-written one, using another learned classifier. To train both types, the authors used a basic cross-entropy loss (for coarse) and contrastive learning (for finer). Especially, they used contrastive learning to make the detector learn how to discriminate humanized texts with two axes: (1) different levels of humanized contexts and (2) different methods of humanization. They ran several experiments on HC3/SeqXGPT-Bench/CheckGPT, and found that their method outperforms the other methods, especially on detecting type 2 (finer type). Also, they showed that their contrastive learning method can be successfully adopted to improve previous methods with patching experiments. Furthermore, they showed that their method can discriminate newly proposed humanization methods, using masked/unseen humanization experiment.

**Strengths:**

- A high-performing two-stage training method for detecting AI-generated texts and its humanized version.
- Detailed experiments on whether the method can be applied to near-real situations, such as unseen humanized methods.

**Weaknesses:**

- Writing is hard to follow, because similar terms are overused; for example, line 98, the authors stating "(1) AI texts humanized in the same method, (2) AI texts with humanized methods of the same level, (3) humanized AI texts and original AI texts, (4) human texts and AI texts" which are very confusing. I understand that the authors had to use terms like "humanized" and "human," but a reader might need some assistance to clearly distinguish those terms (such as tables). I think that those terms can be clearly distinguished using two axes: humanized-level and humanized-methods.
- Need some clarification on the models and experimental methods, to enhance the reproducibility and understanding of a reader. In a similar vein with the first weakness, it is not clear how those methods are used for inference setting. I think the authors assumed that the inference setup receives a single unknown text, but as the current section put much emphasis on contrastive learning with multiple examples, a reader might be confused with those points.

**Questions:**

## Question A. Presentation

I think the authors should provide more detail in the main manuscript, to help readers easily understand and discover possible limitation of the work. There are some missing details which critically affect the reproducibility of the paper.

A1. Are those functions $score$ and $scores$ different in Equation 2? Also, how did you measure $score$?

A2. What is $s_i$'s in equation 4? I think those indicates different items in a single batch, but I don't understand why there are 12 $s_i$s.

A3. What is $\mathcal{L}_{l-human}$ in equation 7?

A4. The method says that it is trainable, but I cannot see any trainable parameters specified in the equations, except text encoder $\Phi$. Then, which text encoder did the authors use? Which characteristic should be satisfied to train the model? I know that the authors stated it in Appendix G. But it is unclear that how did the authors trained the scoring function for Type 1 without updating the parameters of $\Phi$, because the specification about $score$ function is missing.

A5. HC3 and other datasets does not provide humanized text in general. Then, how did the authors collected humanized texts? As those humanized texts can affect the quality and detectability of AI-generated texts, this should be clearly stated to make the experiment more reproducible. Although Appendix D specifies the related points, I think it is not sufficient to recover the whole work.

A6. Task definition is somewhat unclear. In task 1, humanized texts are regarded as human text. But, in task 2, humanized texts should be filtered from human-written text. So, does the authors regard humanized text as human text, or is the reverse true? I understand that the authors want to classify humanized text as human in task 1 because human modified the AI-generated one, such distinction makes me confused when I first read the paper. Clear distinction between "human" text and "humanized" text is needed to make the point of the paper clearer.

## Question B. Discussion

B1. Were there any differences between humanization levels? For example, did those texts humanized in paragraph level show much lower performance than others? Such discussion might provide some valuable insights whether contrastive learning can be applied regardless of the levels.

B2. Did the authors mix multiple humanization methods when generating the humanized text? After reading "hybrid humanizing attacks", it seems that the authors didn't consider such cases in results of Section 4.2.
B2-1. As a follow-up question, does a hybrid of multiple attack makes the detection harder?

## Question C. Related works (Motivations)

C1. In "multi-stage detection" of related works section, the authors mentioning multi-stage works from other tasks or fields. It seems irrelevant though the authors tried to provide the root of their method. So, are there any other similar approaches (in a broad sense) within AI-generated text detection?
C1-1. Can't we use two different models for detecting two tasks? That is, what about using the same model architecture (e.g., SeqXGPT) for both tasks? Why do we need different approach compared to those?

---

### Official Review · Reviewer_YqBm · 2025-11-12

**Soundness:** 2
**Presentation:** 2
**Contribution:** 2
**Rating:** 4
**Confidence:** 3

**Summary:**

This paper introduces HiDet, a hierarchical AI text detector with a coarse module for simple cases and a subdivision module using multi-grained contrastive learning to identify humanized AI texts. Experiments on multiple datasets show HiDet achieves state-of-the-art robustness and can enhance existing detectors as a plug-and-play component.

**Strengths:**

The hierarchical coarse-to-fine design effectively improves robustness against humanized AI texts.

**Weaknesses:**

1. The paper omits comparisons with several strong recent detectors (e.g., DetectLLM, IPAD, Binoculars) that could better situate HiDet among state-of-the-art methods.

2. The additional contrastive module and two-stage inference could increase latency or energy consumption, but inference-time complexity and scalability are not analyzed.

3. Paper clarity needs improvement, e.g. dataset construction details for humanized samples.

**Questions:**

What is the computational overhead of running the two modules sequentially in large-scale or real-time scenarios? Is there an overhead benefit over other detection methods?

---

### Note · Authors · 2025-11-19

I have read and agree with the venue's withdrawal policy on behalf of myself and my co-authors.